# Diurnal Interplay between Epithelium Physiology and Gut Microbiota as a Metronome for Orchestrating Immune and Metabolic Homeostasis

**DOI:** 10.3390/metabo12050390

**Published:** 2022-04-26

**Authors:** Juan Jose Martínez-García, Dominique Rainteau, Lydie Humbert, Antonin Lamaziere, Philippe Lesnik, Mathias Chamaillard

**Affiliations:** 1Laboratory of Cell Physiology, INSERM U1003, University of Lille, F-59019 Lille, France; juan-jose.martinez-garcia@inserm.fr; 2Centre de Recherche Saint-Antoine, CRSA, AP-HP.SU, Hôpital Saint Antoine, Département de Métabobolomique Clinique, Sorbonne Université, INSERM, F-75012 Paris, France; dominique.rainteau@sorbonne-universite.fr (D.R.); lydie.humbert@sorbonne-universite.fr (L.H.); antonin.lamaziere@sorbonne-universite.fr (A.L.); 3Institut National de la Santé et de la Recherche Médicale (INSERM, UMR_S 1166-ICAN), Sorbonne Université, F-75012 Paris, France; philippe.lesnik@upmc.fr; 4Research Institute of Cardiovascular Disease, Metabolism and Nutrition, Faculté de Médecine—Hôpital Pitié-Salpêtrière, F-75013 Paris, France

**Keywords:** aging, antimicrobial peptides, bile acids, circadian rhythms, diurnal oscillations in gut microbiota composition, intestinal epithelial cells, mucus, short-chain fatty acids

## Abstract

The behavior and physiology of most organisms are temporally coordinated and aligned with geophysical time by a complex interplay between the master and peripheral clocks. Disruption of such rhythmic physiological activities that are hierarchically organized has been linked to a greater risk of developing diseases ranging from cancer to metabolic syndrome. Herein, we summarize the molecular clockwork that is employed by intestinal epithelial cells to anticipate environmental changes such as rhythmic food intake and potentially dangerous environmental stress. We also discuss recent discoveries contributing to our understanding of how a proper rhythm of intestinal stem cells may achieve coherence for the maintenance of tissue integrity. Emerging evidence indicates that the circadian oscillations in the composition of the microbiota may operate as an important metronome for the proper preservation of intestinal physiology and more. Furthermore, in this review, we outline how epigenetic clocks that are based on DNA methylation levels may extensively rewire the clock-controlled functions of the intestinal epithelium that are believed to become arrhythmic during aging.

## 1. Introduction

In 1943, The German theoretical biologist Adolf Meyer-Abich introduced the holobiont, a concept that refers to symbiotic associations throughout a significant portion of an organism’s lifetime (from the Greek word holos, meaning whole or entire). This concept is now applied to the studies of host–microbiota interactions [1]. Although speculative, it is now conceivable that the adaptation of gut microbial communities to their nutrient-rich environment had contributed to the host’s overall fitness over millions of years of coevolution. Recent progress in high-throughput sequencing has given us a detailed view of the exquisitely balanced architecture of the microbiota. Among environmental cues that may affect the composition of the gut microbiota from the earliest days of life, daily alternations in light may coincide with changes in air temperature, food availability and humidity, suggesting a potential advantage for anticipating and adapting to such changing environmental conditions that operate as zeitgebers [2]. The circadian clock is an endogenous, time-tracking system that generates self-sustaining oscillations in a hierarchical manner (from the Latin circa diem, which means about a day). A plethora of physiological functions from several subordinate organs are entrained by the circadian system to a phase and a time period that is close to the 24 h solar day. Among internal timekeeping processes that were acquired early in evolution, several photic and non-photic cues influence biological behavior and the organismal physiological state on a daily basis through a coordinated communication between the peripheral clocks and central pacemaker located in the hypothalamic suprachiasmatic nucleus (Figure 1). Specifically, the latter is specialized in relaying temporal signals to the peripheral clocks through the neuronal and humoral pathways for maintenance of a neurological and metabolic synchrony across the sleep–wake cycle. This interorgan communication in response to the light/dark cycle is able to orchestrate a complex sequence of events that includes a fascinating regulation of the stability of several molecular timekeeping mechanisms, including the stability of the gut microbiota composition. A dysregulation of all natural phenomena that exhibit a diurnal rhythm over the course of the day has been linked to a myriad of pathological processes. The purpose of this review is to discuss recent advances that have contributed to our understanding of how specific functions of intestinal epithelial cells oscillate within a period of 24 h. Notably, we discuss how daily food ingestion may rewire several functions of intestinal epithelial cells and their subsequent impact on the regulation of biological processes at a systemic level [3,4]. Among signals that communicate this information from intestinal epithelial cells to the rest of the body, it is becoming more and more clear that the epithelial interaction with metabolites from the gut microbiota acts as a relay for maintenance of intestinal homeostasis and other systemic physiological processes, including cholesterol metabolism and bile acid synthesis.

## 2. The Clock Machinery of Intestinal Epithelial Cells Is a Relay for Shaping Circadian Homeostasis

Daily fluctuation observed at the cellular level implies a complex interaction of factors that are intrinsic to intestinal epithelial cells. Even at a local level, many aspects of biological behavior and physiology of intestinal epithelial cells are temporally controlled, including drug detoxification, barrier function, the bile acid enterohepatic cycle, antimicrobial defense and intestinal peristalsis (Figure 2). As another example, time-of-day variation in the renewal of the epithelium coincides with diurnal changes in the composition of the gut microbiota. Such a fluctuation observed at the cellular level implies a complex interaction of factors that are either extrinsic or intrinsic to intestinal epithelial cells. At the molecular level, the circadian clock network consists of transcription–translation feedback loops (TTFLs). *Bmal1* binds to *Clock* in the nucleus, and the activated heterodimer induces the expression of several genes containing E-box in the promoter site. Some of these genes are *Per1*, *Per2*, *Per3*, *Cry1*, *Cry2*, *Nr1d1* (that is encoding for Rev-erb alpha), Ror alpha and *Dbp*. *Per* and *Cry* are then able to translocate to the nucleus and regulate the Bmal1 activity. Another level of molecular complexity is that Rev-erb alpha and Ror alpha are involved in the repression of *Bmal1* activity. A third level of regulation is governed by *Dbp* and the bZIP transcription factor NFIL3 (nuclear factor interleukin-3-regulated), which regulates Per expression by activation or inhibition. These clock genes, once expressed, regulate the expression of other genes that directly control biological processes; these are termed the clock-controlled genes (CCGs) [5]. Bmal1 expression is controlled by a positive and a negative loop in intestinal epithelial cells induced by NFIL3 and SIRT1, respectively. Interestingly, NFIL3 has been discovered to confer a protection against obesity by diminishing the STAT3 pathway and consequently diminishing fatty acid synthesis. NFIL3 expression is controlled by Rev-erb alpha. Contrarily to NFIL3, SIRT1 induces insulin resistance and inhibits catabolism in response to a high activation of Bmal1/Clock dimer [6,7]. This was demonstrated in studies using Bmal1-deficient mice floxed with VillinCRE as an intestinal epithelial-specific deficient model of the circadian clock. In this work, Bmal1 deficiency diminished Mrp2 mRNA and protein levels, activated by *dbp* and repressed by *E4bp4*. Furthermore, *Bmal1* directly activated the transcription of dbp and Rev-erb alpha and negatively regulated *E4bp4* and *Mrp2* by Rev-erb alpha [8]. For a more complete view of these notions on additional peripheral clocks, we refer the reader to an excellent review by Koronowski and Sassone-Corsi [9]. In some instances, such an impairment of circadian processes can subsequently increase susceptibility to disease such as metabolic syndromes and obesity through numerous transcription/translation oscillation loops and epigenetic changes at one time of the day relative to another [10]. Several questions now arise regarding how either light deprivation or ambient light pollution may negatively impact peripheral tissues that are not directly sensitive to light, such as the intestinal epithelial barrier.

## 3. The Clock Machinery Intrinsically Coordinates the Proliferation of Intestinal Stem Cells and the Fate of Their Daughter Cells

Under normal conditions, diurnal fluctuations in the proliferation of intestinal stem cells have been noticed within the crypt for some time [11]. The sensitivity of intestinal epithelial cells to oxidative damage has been linked to circadian oscillations [12]. From a molecular point of view, *Per2* reduces cell cycle progression and cellular proliferation downstream of the beta-catenin pathway in intestinal epithelial cells [13]. *Per2* gene silencing was found to enhance cell proliferation and reduce cellular apoptosis in isolated epithelial cells [14]. Furthermore, it is worth noting that attenuation of *Bmal1* function resulted in downregulation of genes in the canonical Wnt pathway [15]. The amplitude of the cyclic variation is also enhanced in terminally differentiated intestinal epithelial cells that have irreversibly lost their ability to proliferate when compared to the dividing intestinal stem cells and Paneth cells that reside at the crypt base [16]. Such control of the self-renewal capacity is maintained by several Paneth-cell-secreted Wnt ligands that indirectly influence the gut microbiota composition. This implies a bidirectional interplay, as the clock-controlled expression of several regulators of the cell cycle (including the orphan nuclear receptor retinoic-acid-related orphan receptor alpha (ROR alpha)) orchestrates mutualistic interactions with the microbiota [17]. Studies on stem cells demonstrated that the expression of several circadian genes is repressed by an intrinsic program that is induced during their differentiation process [18,19]. Besides those studies in mammals, other studies in *Drosophila* sp. revealed that intestinal stem cells do not necessarily require clock genes during differentiation. In one such study, a generated GFP-*per* reporter gene demonstrated a high heterogeneity of the activity of the circadian clock machinery among discrete subsets of intestinal epithelial cells, and it was not present in enteroendocrine cells [20]. Nevertheless, the use of the murine fibroblast cell line NIH3T3 cells corroborated the concept that circadian clocks, mediated by a highly robust synchronized expression and/or repression of circadian genes between two oscillators, contribute to the dividing capacity of stem cells [21]. Although challenging, further work is required to properly understand how environmental cues are involved in the metabolic reprogramming of stem cells to meet their functional bioenergetic needs at any time of the day.

## 4. Cyclic Variations in the Functionality of Secretory Intestinal Epithelial Cells and Interactions with the Gut Microbiome

Over the past twenty years, it has been documented that immune reactions fluctuate in magnitude to anticipate the daily time window of greater likelihood of infection. Interestingly, it has recently emerged that the abundance of several discrete subsets of bacteria is submitted to circadian oscillations [22]. The gut is inhabited by a large number of microorganisms with variable needs and behaviors, and it is now becoming clear that circadian rhythms regulate the composition of the intestinal microbiota through several mechanisms to be discussed below. Preliminary studies showed that gut hormones, glucocorticoids and serotonin are secreted by intestinal epithelial cells in a circadian manner [23,24]. Emerging work from the Weizmann Institute of Science has provided evidence that the dominant takeover of a dysbiotic microbiota is properly regulated by the epithelial secretion of antimicrobial peptides (namely, angiogenin-4, Intelectin 1 and Resistin-like molecule β/FIZZ2) downstream of IL-18 signaling [25]. There is growing appreciation, as a result of transcriptomic and epigenetic studies, of the importance of the diurnal fluctuations of bacterial attachment to the epithelial surface as instrumental for temporally modulating the functional outcome of oscillating transcriptional and epigenetic programs as a way to anticipate potential threats, including bacterial infection [26]. Furthermore, the dysbiotic microenvironment increased susceptibility to the intestinal colonization of pathobionts, such as *Salmonella*. Recently, a study revealed that bacterial attachment triggers a STAT3-dependent antimicrobial response to daytime feeding changes in intestinal epithelial cells [27]. Mechanistically, the authors identified that the attachment of SFB is needed for optimal induction of a STAT3-dependent gene program in intestinal epithelial cells during the feeding period. Consequently, a greater susceptibility to *Salmonella* infection was noticed as a consequence of the adherence of SFB that concomitantly leads to an overproduction of Reg3 gamma by intestinal epithelial cells. However, vancomycin treatment lowers the production of Reg3 gamma in models of colitis, leading to greater fungal fitness. The authors provided convincing evidence that such a protective effect of fungi on the severity of colitis is caused by an increase in the abundance of Proteobacteria, especially *Enterobacteriaceae* [28]. When considering this aspect, future investigation will be needed to determine the extent to which the clock-mediated regulation of antimicrobial peptide secretion by intestinal epithelial cells is similarly important for responses against other microorganisms, such as fungi and enteric viruses. Establishing all of the consequences for organismal output will be key to further chronopharmacological investigation to improve the efficacy of immune surveillance, including those that also contribute to defense against cancers.

## 5. Epithelium Integrity Is Intrinsically Coordinated by the Clock Machinery of Intestinal Epithelial Cells

Besides defense response to oxidative stress, the colonic permeability of ions, nutrients, and water is a mere output function of the circadian system in intestinal epithelial cells. Specifically, the expression of occludin and claudin-1 exhibits daily variations that are consistent with the nighttime nadir of cortisol. A recent study provided evidence that the key clock genes Period1 (*Per1*) and *Per2* were expressed in antiphase with the aforementioned tight-junction proteins. It should be noted that the expression of occludin and claudin-1 is constantly lowered in the colon of mice bearing a mutation in the *Per2*-encoding gene, which were more resistant to colonic injury induced by dextran sodium sulfate (DSS) than wild-type mice [29]. *Per2* acts as an antisense oscillator and directly contributes to the repression of clock-controlled target genes through several mechanisms. Notably, the nucleocytoplasmic shuttling of *Cry1/2* is regulated by *Per2*, which rhythmically interacts with several RNA-binding proteins that may control the expression of tight-junction proteins [30]. However, how the circadian clock transcriptionally regulates their expression has not been clarified in enough detail. The loss of the oscillation of *Per2* enhanced expression of several genes involved in epithelial–mesenchymal transition [31]. Such data are of interest, given that the loss of oscillation coincided with lowered levels of genes that are associated with progenitor/epithelial cell function, such as the α6 integrin that mediates the adhesion of epithelial cells to laminin [32]. The authors linked this observation to a mechanical control that mainly depends on the extracellular matrix. The extracellular matrix was described as one of the main actors that communicate with surrounding cells to control their clock biology. Experiments using murine organoids demonstrated that the extracellular matrix then promotes contractile movements in these cells in vitro. This contractile behavior was dependent on the increase in the percentage of oxygen at the surface of intestinal epithelial cells [33]. However, it is currently not clear how the clock machinery in intestinal epithelial cells is molecularly influenced by the stiffness of the extracellular matrix to provide protection against fragility and shedding. Answering this question will require further detailed studies making use of synchronized organoids and mouse models with specific defects of several clock genes in intestinal epithelial cells. Among molecules of interest is PPAR-gamma, loss of which heightens epithelial oxygenation and results in the loss of barrier function through actin disassembly from the cytoskeleton [34]. Although it has not been addressed experimentally, it makes sense to consider the metabolic reprogramming of intestinal stem cells through fatty acid oxidation as instrumental in the divergence of their proliferative capacity when compared to their daughter cells, in which full activation of PPAR-gamma is attained (Figure 2). Furthermore, it is important to keep in mind that the amplitude of clock gene expression is enhanced in soft 3D microenvironments compared to stiff 2D environments [35].

## 6. The Circadian Clock Machinery of Intestinal Epithelial Cells Regulates Their Response to Oxidative Stress

Circadian rhythmicity is reinforced by several nutrients, energy and redox level signals to adapt intestinal physiology to specific needs within the framework of the solar day. Below, we review recent findings on the daily fluctuations in the expression or activity levels that have been measured for many enzymes that protect the intestinal epithelium from oxidative stress [36]. Specifically, intestinal epithelial cells express several genes of the clock machinery, deletion of which has an impact on cellular response to oxidative stress, including xenobiotic/drug detoxification [36]. This was elegantly demonstrated in studies using a model of clock machinery deficiency in intestinal epithelial cells. Loss of the aryl hydrocarbon receptor nuclear translocator-like protein 1 (*Arntl*, also referred to as *Bmal1* for brain and muscle ARNT-like 1) in intestinal epithelial cells lowered the expression of the multidrug-resistance-associated protein 2 (Mrp2), which is required to limit methotrexate absorption by blocking entrance into enterocytes. In everted gut sac experiments, the mucosal-to-serosal transport and intestinal accumulation of methotrexate was enhanced as a consequence of loss of Bmal1 in intestinal epithelial cells [8]. This is of particular importance, as methotrexate is an inhibitor of dihydrofolate reductase, and its anti-inflammatory action relies on the production of reactive oxygen species (ROS) [37]. When reaching elevated intracellular levels of ROS, one may anticipate some cellular damage, including lipid peroxidation. It is conceivable that continuous oxidative stress throughout the day leads to pathological conditions, including leaky gut and fibrosis. However, some bacteria, such as *Lactobacillus paracasei*, transfer lactate to the epithelial cells, in which acetyl-coA synthesis occurs in the presence of oxygen, favoring the tricarboxylic acid cycle (TCA) and fatty acid synthesis [38]. Dissecting how the clock machinery properly regulates the interplay between innate immunity and redox signaling will contribute to the understanding of downstream defense responses when oxidative stress excessively occurs in intestinal epithelial cells. It has been demonstrated in stem cells and epithelial cells from other organs that *Bmal1* can negatively regulate the expression of the mitochondria Ucp1 protein to reduce the amount of ATP generated through an oxidation of fuels [15] and ROS generation [39]. Among several possible explanations, the potential contribution of peroxisome-proliferator-activated receptor delta (PPAR-delta), which is activated upon fasting, remains to be determined [40]. Consistent with this concept, the formation of ketone bodies is induced by long-term fasting and depends on changes in the microbiome. Specifically, increased ketone bodies were observed in conventionalized mice compared to germ-free animals [41]. In addition, the use of the *Arntl*-floxed villin-Cre mice revealed that the epithelial expression of *Bmal1* contributes to obesity development, body weight gain and related abnormalities, such as hyperlipidemia, through decreased lipid absorption. Among the underlying mechanisms, the epithelial loss of *Bmal1* led to a reduced expression of the *Dgat2*-encoding gene, which codifies the enzyme of triacylglycerol synthesis [42]. In this sense, it is unclear how circadian regulation of fatty acid oxidation may be correlated with epithelial renewal capacities of stem cells and whether epigenetic chromatin-modifying enzymes may contribute to this fairly dynamic process at steady state. However, natural protective mechanisms against lipid peroxidation and nitric oxide production are intrinsically controlled by the expression of PPAR-gamma in intestinal epithelial cells. Another possibility involves the orchestration of local and systemic lipid metabolism by a signaling circuit linking the epithelial clock to sensing of the gut-specific microbial components by innate immune cells [6].

## 7. Short-Chain Fatty-Acid-Producing Bacteria Impose a Metabolic Choice to Be Made by Intestinal Epithelial Cells

Some gut bacteria are beneficial due to the end products of their metabolism that they provide to the intestinal epithelial cell. Notably, dietary fibers are metabolized in short chain fatty acids (SCFA) by some anaerobic bacteria. The end products of their metabolism subsequently induce a metabolic reprogramming of colonocytes. One intriguing example among others is that PPAR-gamma activation by butyrate promotes aerobic glycolysis, together with fatty acid oxidation [43]. Epithelial loss of PPAR-gamma enhances nitric oxide production to a similar extent as that observed in response to interferon-gamma. Specifically, inducible nitric oxide synthase (iNOS) was inhibited by PPAR-gamma in response to butyrate to prevent dysbiosis that may predispose to colitis. The pantetheinase VNN1 that is regulated by PPAR-gamma is able to reduce the Warburg effect produced under cellular stress conditions, contributing to metabolic homeostasis. This was demonstrated by Giessner and colleagues when pantetheinase, an inducer of Vnn1 expression, was administered to mice, inducing an increase in the mitochondrial function in tumors [44]. In contrast, colonocytes cannot rely on mitochondrial fatty oxidation under inflammatory conditions but rather on anaerobic glycolysis, as observed in cancer cells. Of particular interest, the cellular metabolic choice of colonocytes was restricted by interferon-gamma, creating an environment in which oxygen supply is excessive, leading to an expansion of facultative anaerobic bacteria, such as *Enterobacteriaceae*. Conversely, the inhibition of PPAR-gamma contributes to a deregulation of glucose metabolism and induces an increase in glucose uptake [45]. To some extent, this is similar to the metabolic reprogramming that is needed to support cell fate and function of myeloid cells [46]. Of equal importance is the epithelial function of the pentanoate receptor GPR41 (also called free fatty acid receptor 3), which promotes inflammation [47], whereas pentanoate administration suppresses the generation of small-intestinal Th17 cells in germ-free mice mono-colonized with segmented filamentous bacteria (SFB) [48]. Furthermore, the G-protein-coupled receptor GPR43 facilitates inflammasome activation in colonocytes when treated with acetate [49]. It is conceivable that other acetate-producing bacteria may help epithelial cells generate acetyl-coenzyme A (acetyl-coA), which is required under intracellular anaerobic conditions to promote fatty acid biosynthesis. Another level of complexity that must be considered is the influence of changes in the gut microbiota composition that may lead to the generation of bacterial fermentation products that have the capacity to upregulate the expression of other genes involved in mitochondrial fatty acid oxidation, such as PPAR-alpha, CD36 and carnitine palmitoyltransferase I [50]. Furthermore, it remains to be determined whether metabolites other than SCFA may also impose a choice to be made by intestinal stem cells and their daughter cells. Among such metabolites, phytate and inositol trisphosphate have been reported to control the expression of intestinal epithelial histone deacetylase 3 (HDAC3) [51], which is recruited rhythmically to chromatin and produces diurnal oscillations in histone acetylation and therefore in metabolic gene expression, including CD36 [52]. Thus, HDAC3 represents a converging epigenetic sensor of distinct metabolites that calibrates rhythmic host responses to diverse microbial signals. Furthermore, the activity of PPAR-alpha is undoubtedly influenced by the daily fluctuation in the abundance of Gram-negative bacteria through the epithelial activation of the rate-limiting enzyme Cyp11a1 in cortisol synthesis. Prior studies using genetically modified mouse models of epithelial deficiency in either PPAR-alpha or glucocorticoid receptor confirmed microbiota-dependent mediation of several clock genes to enable lipolysis and lower insulin levels after a long period of antibiotic treatment [17]. However, it remains unclear whether such oscillating phenomena may reflect an epithelial detachment of bacteria at a specific time of day, as observed when SFB is able to adhere to the epithelium to modulate antimicrobial peptide expression during the sleep-to-wake transition [27].

## 8. Systemic Influence of Epithelial Clock Machinery on the Synthesis of Bile Acids and on Gut Motility

Bile acids are synthesized from cholesterol through the activation of cholesterol 7alpha-hydroxylase (CYP7A1) in the liver. After being conjugated with either glycine or taurine, they are released to the intestinal lumen to promote the epithelial absorption of long-chain fatty acids [53]. The circadian clock can change the expression of some genes that control the synthesis of bile acids, such as the farnesoid X receptor (FXR). FXR activation controls bile acid synthesis in the liver and bile acid secretion in organs by regulating the expression of bile acid transporters in both the liver and enterocytes [54]. Some colonic bacteria are then required to specifically deconjugate conjugated primary bile salts and transform them into secondary bile acids during the fasting period. The deconjugation is catalyzed when the bile salt hydrolase (BSH) is synthesised by a discrete subset of bacteria, including *Lactobacillus, Clostridium, Bacteroides, Enterococcus* and *Bifidobacterium* [55]. In response to vancomycin, postprandial fecal concentrations of secondary bile salts and of fibroblast growth factor 19 (FGF19) were significantly reduced, suggesting reduced BSH activity. Interestingly, a compensatory increase in Gram-negative bacteria coincided with a marked reduction in the abundance of Gram-positive bacteria in response to vancomycin. In agreement with the possibility that vancomycin may enhance bile acid synthesis, an increased level of the primary bile acids cholic acid (CA) and chenodeoxycholic acid (CDCA) was inversely correlated with the abundance of SCFA-producing bacteria [56]. Consequently, peripheral insulin sensitivity was significantly lowered, despite the absence of changes in the plasma level of insulinotropic hormones. Intriguingly, some studies demonstrated that glucagon secretion is reduced in mice that are deficient in carnitine palmitoyltransferase (Cpt1a), a transporter of fatty acids to the mitochondria [57]. Of equal importance, vancomycin decreases the abundance of Gram-positive bacteria while changing gut motility [58]. This effect on intestinal motility is likely dependent on the bacterial sensing capacity of Toll-like receptor 4 (Tlr4) at the surface of intestinal epithelial cells [59]. Further work is required to determine whether the decreased insulin sensitivity is a consequence of the vancomycin-induced decrease in the levels of lithocholic acid (LCA) and deoxycholic acid (DCA) that are sensed by the bile acid membrane receptor TGR5 at the surface of colonic epithelial cells for maintenance of systemic insulin sensitivity. This is of particular importance to clarify the sequence of events leading from changes in the gut microbiota composition to a reduced number of enteroendocrine cells in response to dietary lards [60]. Among the bacteria found more abundantly in response to a high-fat diet, *Bilophila wadsworthia* is a sulfite-reducing pathobiont that subsequently affects the epithelial integrity in *Interleukin-10*-deficient mice [61]. Additional work is awaited to understand how the metabolism of bile acids is influenced by other bacteria, such as *Parabacteroides distasonis* and *Ruminoclostridium* [62]. *P. distasonis* produces secondary bile acids and succinate, which are absorbed by the intestinal cell to produce de novo glucose in response to FXR activation [63].

## 9. Aging

Aging is characterized by a progressive decrease in a wide variety of physiological functions. Some studies have demonstrated that cyclicity and activity of clock genes showed significant age-related alterations, becoming fragmented and diminished in elder individuals. This evidence suggests that there is a progressive degeneration of the structure and cyclicity of the circadian timing that induces a break of the control of the rest and sleep phases in the elderly [64]. Recently, epigenetic events were discovered as a consequence of hypoxia. Specifically, hypoxia-inducible factor 1 alpha (HIF1α) modifies the epithelial expression of the HDAC3 that controls epithelium integrity [65]. These pieces of evidence open the road for additional investigations on how the oscillations of the gut microbiome may occur. This conundrum is particularly difficult to address, as it will require a more detailed understanding of how arrhythmic bacterial adhesion may precisely manipulate the functionality of different cells at a specific time of the day. We can further estimate that the epigenetic events induced by environmental factors during aging are cause a metabolic alteration in intestinal epithelial cells as a consequence of the clock asynchrony. However, further investigations are required to better understand how alterations in the gut microbiome may confer a greater risk of degenerative diseases and cancer later in life as a potential consequence of epigenetic events.

## 10. Perspectives

A wealth of recent studies unveiled that intestinal epithelial cells play an essential role in relaying signals both to and from the intestinal microbiota during either fasting or dietary intake time windows. One may speculate that the diurnal fluctuations in the composition of the gut microbiome help to assist host defense and metabolic homeostasis. From an evolutionary perspective, this concept implies the need for specific molecular passwords that are released at a time when there is a specific need for either anabolism or catabolism. A deregulation of the axis induced by diet, antibiotics or other dysbiosis-promoting factors is challenging for the host and may contribute to disease development upon failure to properly compensate for the functionality of the gut microbiome. Notably, an alteration of the bile acid sensing and transport by intestinal epithelial cells may influence liver metabolism through PPAR-alpha-dependent mechanisms that are beginning to be understood. As an excellent example, it was described how a high-fat diet promotes a dysbiosis environment that alters the clock machinery of peripheral organs, leading to increased permeability and insulin resistance [66]. Furthermore, a high-fat diet can induce a disruption of the circadian oscillation of some important genera with BSH activity, such as *Lactobacilli* [67]. The importance of the circadian regulation of feeding time for the microbiota was demonstrated by the application to *db/db* mice of an intermittent fasting of 24 h; this condition was compared with ad libitum fed diabetic mice. During the intermittent fasting, microbiota composition of *db/db* mice was abruptly changed, increasing the abundance of *Lachnospiraceae*, *Lactobacillus, Oscillospira, Ruminococcus* and *Clostridiales* at the fecal level. In contrast, the abundance of *Akkermansia* and *Bacteroides* decreased in obese mice with intermittent fasting [68]. It is now evident that epithelial cells play a key role to control alterations of the gut microbiome. However, it remains difficult to mechanistically characterize the advantages or disadvantages that each oscillating bacterium may induce individually or collectively at either a steady state or upon metabolic stresses. Additional studies are required to optimize the design of drug dosing regimens for individuals with a disrupted circadian clock who may experience a lowered drug detoxification, leading to defects in ROS scavenging and/or antioxidant signaling.

## Figures and Tables

**Figure 1 metabolites-12-00390-f001:**
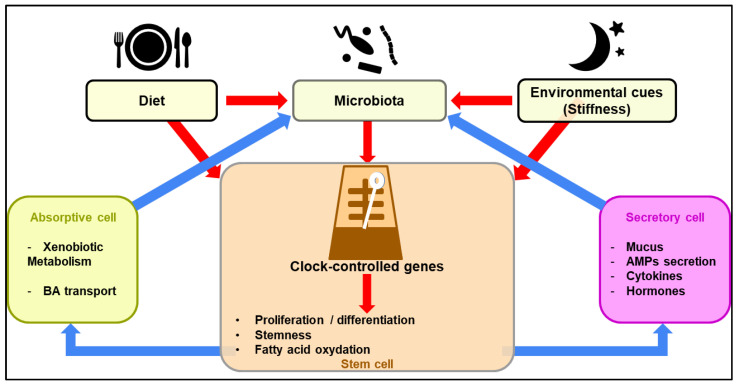
Diagrammatic overview of the regulation of clock-controlled function of intestinal epithelial cells upon sensing of environmental cues. Diurnal fluctuations in the gut microbiota composition may regulate the ability of stem cells to proliferate and differentiate through a complex metabolic reprogramming (red arrows). This leads to a concerted action by intestinal epithelial cells on the gut microbiota through both positive and negative feedback loops (blue arrows).

**Figure 2 metabolites-12-00390-f002:**
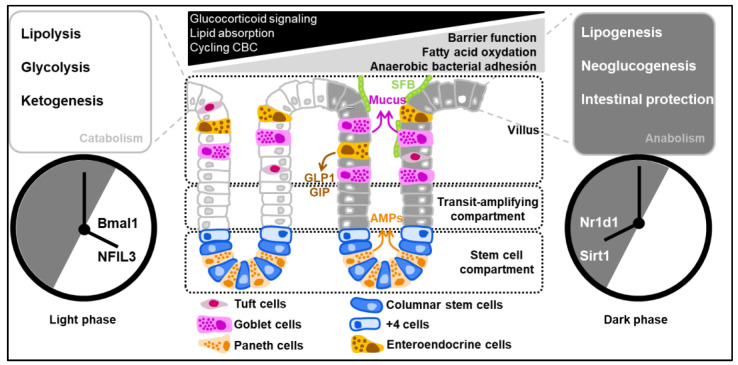
Diurnal oscillations of the clock-controlled genes regulate the catabolism and anabolism of intestinal epithelial cells. Clock-controlled genes are regulated by a complex interplay of positive and negative loops within the framework of the solar day. As an example, Period 2 is thought to be repressed by the bZIP transcription factor NFIL3 that is negatively regulated when the expression of *Nr1d1* (the gene that codifies Rev-erb alpha) reaches its zenith.

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
