# Peer review of "Diurnal Interplay between Epithelium Physiology and Gut Microbiota as a Metronome for Orchestrating Immune and Metabolic Homeostasis"

_metabolites, 2022, doi:10.3390/metabo12050390_

Round 1
Reviewer 1 Report
The interplay between the epithelium and the gut microbiota as a metronome for orchestrating intestinal physiology and beyond
By JJ Martinez-Garcia et al (Corresponding author: M Chamaillard)
Submitted to Metabolites (Editorial No. metabolites-1648872)
General Comments
This review is a bold attempt to describe the interrelationship between the gut epithelia and their physiology including diurnal molecular clocks and the growth and various functions of the gut microbiota. The recent publications on this topic are simultaneously fascinating and very complex.
The submitted manuscript is rather multifaceted and not always clear.
For improvement, the following sections are suggested:
- Physiology and pathophysiology of gut epithelium
- Circadien clock machinery of gut epithelia
- Gut microbiota and relationship to host
- the concept of holobiont
- bacterial gut fermentation products and host metabolism
- importance of diet for gut microbiome composition
- eubiosis, dysbiosis and disease
- immune responses
- intestinal microbiome-derived and host metabolites (combined metabolome)
- future research
Suggested additional refs
van de Guchte M, Blottière HM, Doré J. Humans as holobionts: implications for prevention and therapy. Microbiome. 2018 May 1;6(1):81. doi: 10.1186/s40168-018-0466-8. PMID: 29716650; PMCID: PMC5928587.
Koh A, Bäckhed F. From Association to Causality: the Role of the Gut Microbiota and Its Functional Products on Host Metabolism. Mol Cell. 2020 May 21;78(4):584-596. doi: 10.1016/j.molcel.2020.03.005. Epub 2020 Mar 31. PMID: 32234490.
Krautkramer KA, Fan J, Bäckhed F. Gut microbial metabolites as multi-kingdom intermediates. Nat Rev Microbiol. 2021 Feb;19(2):77-94. doi: 10.1038/s41579-020-0438-4. Epub 2020 Sep 23. PMID: 32968241.
Debnath N, Kumar R, Kumar A, Mehta PK, Yadav AK. Gut-microbiota derived bioactive metabolites and their functions in host physiology. Biotechnol Genet Eng Rev. 2021 Oct;37(2):105-153. doi: 10.1080/02648725.2021.1989847. Epub 2021 Oct 22. PMID: 34678130.
Rodriguez DM, Benninghoff AD, Aardema NDJ, Phatak S, Hintze KJ. Basal Diet Determined Long-Term Composition of the Gut Microbiome and Mouse Phenotype to a Greater Extent than Fecal Microbiome Transfer from Lean or Obese Human Donors. Nutrients. 2019 Jul 17;11(7):1630. doi: 10.3390/nu11071630. PMID: 31319545; PMCID: PMC6682898.
Harris VC, Haak BW, Boele van Hensbroek M, Wiersinga WJ. The Intestinal Microbiome in Infectious Diseases: The Clinical Relevance of a Rapidly Emerging Field. Open Forum Infect Dis. 2017 Jul 8;4(3):ofx144. doi: 10.1093/ofid/ofx144. PMID: 28852682; PMCID: PMC5570093.
Giovanni MY, Schneider JS, Calder T, Fauci AS. Refocusing Human Microbiota Research in Infectious and Immune-mediated Diseases: Advancing to the Next Stage. J Infect Dis. 2021 Jul 2;224(1):5-8. doi: 10.1093/infdis/jiaa706. PMID: 33188418; PMCID: PMC8253126.
Vlasova AN, Takanashi S, Miyazaki A, Rajashekara G, Saif LJ. How the gut microbiome regulates host immune responses to viral vaccines. Curr Opin Virol. 2019 Aug;37:16-25. doi: 10.1016/j.coviro.2019.05.001. Epub 2019 Jun 1. PMID: 31163292; PMCID: PMC6863389.
Desselberger U. Differences of Rotavirus Vaccine Effectiveness by Country: Likely Causes and Contributing Factors. Pathogens. 2017 Dec 12;6(4):65. doi: 10.3390/pathogens6040065. PMID: 29231855; PMCID: PMC5750589.
Parker EP, Ramani S, Lopman BA, Church JA, Iturriza-Gómara M, Prendergast AJ, Grassly NC. Causes of impaired oral vaccine efficacy in developing countries. Future Microbiol. 2018 Jan;13(1):97-118. doi: 10.2217/fmb-2017-0128. Epub 2017 Dec 8. PMID: 29218997; PMCID: PMC7026772.
Desselberger U. Significance of the Gut Microbiome for Viral Diarrheal and Extra-Intestinal Diseases. Viruses. 2021 Aug 12;13(8):1601. doi: 10.3390/v13081601. PMID: 34452466; PMCID: PMC8402659.
The Figures 1 and 2 should be improved and carefully constructed explanatory legends be provided.
A list (alphabetical) of the multiple abbreviations used should be provided at the bottom of p1.
Specific Comments
Line
2 Reconsider title, e.g. ‘Intestinal cell physiology and gut microbiota: structural and functional interrelationships’, or similar.
19 Consider phrasing: … metabolic syndromes.
25f Clarify the meaning of the sentence.
30 Replace ‘Prelude’ by ‘Introduction’.
45 Fig. 1. Consider phrasing: ‘Diagrammatic overview of relationships among host cells including molecular clocks, gut microbiome, diet, combined host-microbiome metabolites’. A carefully worded explanatory legend is required.
51 … to an excellent review [1].
74 Fig. 1. See comment line 45.
81 Provide details and relevant refs on oxidative stress.
92 Provide relevant refs.
104f Consider reading: … Among several possible explanations, the potential contribution of… upon fasting remains to be determined.
110 … contributes to obesity development…
115 … may be correlated with…
124 … Some gut bacteria…
127 Provide relevant refs.
133 … to prevent dysbiosis that may predispose to colitis… Please clarify and provide relevant refs.
145 Provide a relevant ref for the statement.
153 … to promote fatty acid biosynthesis…
172 … specific time of the day as observed when…
183 Spell out at first mentioning. See general comment on abbreviations.
187 … tight junction proteins [24]. However, how …
202 … mouse models…
205ff Clarify the meaning of the sentence.
209 and 230. Figure 2. Rephrase the heading and provide a carefully worded explanatory legend.
236 … is expressed to higher levels…
240 and 245. Explain WNT/wnt. Decide on orthography.
251f Clarify the meaning of the sentence.
258 Although challenging, further work is required…
261 Consider phrasing: Cyclic variations … epithelial cells and interaction with the gut microbiome.
268 … to be discussed below.
271 … properly regulated by the epithelial secretion…
279 … Recently a superb study revealed that…
287 … The authors provided convincing evidence that… caused by an increase in concentration of Proteobacteria…
290 … needed to determine…
303 … The circadien clock can change…
318 … Consequently, … Please explain.
323 … mitochondria [49]. Vancomycin decreases … while changing gut motility.
326 and 336. … Further work is required…
341 … One may speculate… homeostasis. From an evolutionary perspective…
346 … may confer a greater risk later in life. … Please clarify.
352 Only recently some epigenetic events have been discovered as a consequence…
356 … stiffness of the matrix… Please clarify.
359 … diet, antibiotics, or other dysbiosis promoting factors…
362f This sentence is cryptic.
Reviewer 2 Report
The review aims to compile information about the circadian rhythms in intestinal epithelial cells (IECs) and how they can be modulated by microbiota. There are some considerations that could be improved:
- The subsections could be re-ordered placing 4 and 5 at the beginning as those partially explain the composition of the intestinal mucosa and IEC types (information regarding these general intestinal/IEC biology could be better addressed). A section explaining how the composition of the microbiota changes over the day and differences in adherence could be added afterwards. Also, some sections do not seem to properly address the connection between IEC and microbiota.
- Figure 2 legend seems incomplete (e.g. what is the green item?), and leading genes for the main pathways could be specified. There is no clear explanation of this figure in the figure legend or in the text, i.e. the IEC subtypes highlighted in the figure are not always mentioned or explained in the review and the figure does not show all IEC subtypes.
- Figure(s) depicting the main IEC-microbiota interactions highlighted in the review will help understand the subject.
- Some concepts or examples could be specified better: the clock genes that are expressed in IECs (lines 81-82), Warburg effect (line 134).
- Closing remarks on each part of the review is sometimes long and comments belong to the concluding perspectives section. The perspective section is currently too long and ageing could be a separate subsection.
Round 2
Reviewer 1 Report
The diurnal interplay between the epithelium physiology and the gut microbiota as a metronome for orchestrating intestinal homeostasis and beyond
By JJ Martinez Garcia et al (Corresponding author: M Chamaillard)
Submitted to Metabolites (Editorial No. metabolites – 1648872R1)
General Comments
This is the revised version (R1) of a manuscript, the original submission of which has been studied and commented upon by this reviewer. The authors have considered the comments/suggestions carefully and followed many of them. The R1 version has considerably improved. Since the findings reviewed are very complex, it should be attempted to intersperse the text here and there with concluding remarks. It should also be considered to cite/assess the following previously mentioned refs: Debnath et al, 2019; Rodriguez et al, 2019; Giovanni et al, 2021; Desselberger, 2021.
Specific Comments
Line
2 The authors have changed the title in a way which is rather convoluted. Consider:
‘The diurnal interplay between the epithelial metabolism and the microbiota of the gut in homeostasis and dysbiosis’, or similar. [… The meaning of ‘and beyond’ is unclear.]
26 Phrase: … clocks that are based on…
44 Graphic Abstract, Legend line 2: … epithelial cells that are controlled…
53 … of the gut microbiota from …
74 … within a period of 24 hours…
80 Fig. 1, Legend. Explain blue and red arrows.
87 Fig. 2 is mentioned here first, but only shown after line 324. The figure should be moved up to a space after line 118.
105 … reverb-alpha… [?? Spell out in uniform way throughout the ms]
111 Ref. Yu et al, 2019 is not in the ref. list. Please add and renumber refs in text and list as appropriate.
216 … and consequently the loss of barrier function…
274 … due to the end products of their metabolism…
295 Add the ref. in the list. See comment line 111.
Charo IF. Macrophage polarization and insulin resistance: PPARgamma in control. Cell Metab. 2007 Aug;6(2):96-8. doi: 10.1016/j.cmet.2007.07.006. PMID: 17681144.
324 Fig. 2. Read: ‘Lipogenesis’ in upper right corner.
427ff References.
The following refs are incomplete: 4, 7, 21, 26, 30, 32, 33, 35, 40, 43, 45-47, 57, 58, 62.
Ref. 9. The initials should follow surnames.
Ref. 14. Read: Matsuura T, …
Author Response
Reviewer response point by point:
Includding references:
Debnath et al, 2019; Rodriguez et al, 2019; Giovanni et al, 2021; Desselberger, 2021.
Author response. We thank the reviewer for his/her references inclusion proposal. We have considered the four references. We agree to include the review by Debnath et al. 2019 that provides a useful overview for the readers on the properties on gut-microbiota derived bioactive metabolites in host physiology. However, the remaining three references do not cover circadian clock aspects of host microbiota interaction that is the focus of our review.
Specific Comments
Line
2 The authors have changed the title in a way which is rather convoluted. Consider:
‘The diurnal interplay between the epithelial metabolism and the microbiota of the gut in homeostasis and dysbiosis’, or similar. [… The meaning of ‘and beyond’ is unclear.]
Author response. We thank the reviewer for help us to improve the title with more clarity. We will change the title to the following one:
‘The diurnal interplay between the epithelium physiology and the gut microbiota as a metronome for orchestrating immune and metabolic homeostasis.’
26 Phrase: … clocks that are based on…
Author response. The phrasing has been revised accordingly.
44 Graphic Abstract, Legend line 2: … epithelial cells that are controlled…
Author response. The phrasing has been revised accordingly.
53 … of the gut microbiota from …
Author response. The phrasing has been revised accordingly.
74 … within a period of 24 hours…
Author response. The phrasing has been revised accordingly.
80 Fig. 1, Legend. Explain blue and red arrows.
Author response. The phrasing has been revised accordingly.
87 Fig. 2 is mentioned here first, but only shown after line 324. The figure should be moved up to a space after line 118.
Author response. The figure has been moved accordingly.
105 … reverb-alpha… [?? Spell out in uniform way throughout the ms]
Author response. The phrasing has been revised accordingly.
111 Ref. Yu et al, 2019 is not in the ref. list. Please add and renumber refs in text and list as appropriate.
Author response. The reference has now been added accordingly.
216 … and consequently the loss of barrier function…
Author response. The phrasing has been revised accordingly.
274 … due to the end products of their metabolism…
Author response. The phrasing has been revised accordingly.
295 Add the ref. in the list. See comment line 111.
Charo IF. Macrophage polarization and insulin resistance: PPARgamma in control. Cell Metab. 2007 Aug;6(2):96-8. doi: 10.1016/j.cmet.2007.07.006. PMID: 17681144.
Author response. This additional reference has now been added accordingly.
324 Fig. 2. Read: ‘Lipogenesis’ in upper right corner.
Author response. The phrasing has been revised accordingly.
427ff References.
The following refs are incomplete: 4, 7, 21, 26, 30, 32, 33, 35, 40, 43, 45-47, 57, 58, 62.
Ref. 9. The initials should follow surnames.
Ref. 14. Read: Matsuura T, …
Author response. According to the ref 14, after several bibliographic searches the name of this author is Toru (name) Matsu-ura (surname), currently assistant professor of the Kansai Medical University. The remaining references have now been carefully reviewed accordingly.
